# The Metabolic Aftershock: COVID-19 and Metabolic Disease Risk Among U.S. Active-Duty Military Personnel

**DOI:** 10.3390/metabo15120795

**Published:** 2025-12-14

**Authors:** Kyle W. Sexton, Zella Berill, Apryl Susi, Jacob Coene, Kristan E. Madison, Cade M. Nylund

**Affiliations:** 1Department of Pediatrics, Tripler Army Medical Center, Honolulu, HI 96701, USA; kyle.w.sexton3.mil@health.mil; 2Department of Pediatrics, Uniformed Services University of the Health Sciences, Bethesda, MD 20814, USAapryl.susi.ctr@usuhs.edu (A.S.); kristan.madison@usuhs.edu (K.E.M.); 3Henry M. Jackson Foundation for the Advancement of Military Medicine, Bethesda, MD 20817, USA; 487th Medical Group, Department of Pediatrics, Joint Base McGuire-Dix-Lakehurst, NJ 08641, USA

**Keywords:** COVID-19, type 2 diabetes mellitus, hypertension, hyperlipidemia, metabolic dysfunction-associated steatotic liver disease (MASLD), metabolic syndrome

## Abstract

**Background:** The post-acute sequelae of SARS-CoV-2 infection represent a growing public health concern. While associations between COVID-19 and metabolic disorders have been reported, less is known about this risk in young, healthy populations. This study aimed to quantify the risk of developing new-onset metabolic disorders following SARS-CoV-2 infection among U.S. active-duty service members. **Methods:** We conducted a propensity score-matched cohort study using U.S. Military Health System data between July 2020 and June 2021 of active-duty service members (ADSM) aged 18 to 65 years old. COVID-19 infections were identified through ICD-10 codes or laboratory results. A 1:2 matched cohort compared 103,789 COVID-19 exposed individuals with 207,578 controls. Outcomes included incident diagnoses of type 2 diabetes mellitus (T2DM), hypertension (HTN), hyperlipidemia (HLD), metabolic dysfunction-associated steatotic liver disease (MASLD), and metabolic syndrome (MetS) within one year. Cox proportional hazards models calculated adjusted hazard ratios (aHR) while controlling for obesity and overweight status. **Results:** The median age for both those with and without COVID-19 infection was 26 years (interquartile range 22–33 years), with males comprising the majority of participants (81.1% male, 18.9% female). COVID-19 infection was associated with significantly increased hazards for incident HTN (aHR 1.09; 95% CI, 1.01–1.18), HLD (aHR 1.30; 95% CI, 1.10–1.54), and MASLD (aHR 1.36; 95% CI, 1.15–1.60). However, no significant increased risk was observed for T2DM or MetS. **Conclusions:** COVID-19 infection was associated with significantly increased risk of developing HTN, HLD, and MASLD, highlighting important long-term metabolic consequences in a young, healthy population.

## 1. Introduction

The enduring impact of SARS-Cov-2 infections extends beyond the acute phase, with a growing body of evidence highlighting a significant potential for long term morbidity [1,2]. Notably, a constellation of metabolic sequelae, including obesity, cardiovascular disease and hyperlipidemia, has emerged as a common consequence of COVID-19 infection [1,2,3,4,5,6]. Observational studies have further indicated a concerning rise in the incidence of Type 2 diabetes mellitus (T2DM) during the COVID-19 pandemic, with a strong correlation observed between direct SARS-CoV-2 infection and new-onset diabetes [1,2,3,4,5,6]. Intriguingly, vaccination prior to acute COVID-19 infection has been linked to a reduced risk of developing T2DM and other long-COVID manifestations. While acute COVID-19 and Long-COVID necessitates a spectrum of medical therapies, ranging from target antivirals to systemic corticosteroids, the precise mechanisms underlying the development of new-onset diabetes remains elusive [3,4,5,6,7,8,9,10].

T2DM, a metabolic disorder characterized by insulin resistance and elevated blood glucose levels, is often associated with reduced physical activity and a high comorbidity with obesity [2,4,6]. While acute pancreatic inflammation has been documented in association with other viral infections (e.g., human immunodeficiency virus, mumps, measles, herpes simplex virus, cytomegalovirus virus, hepatitis virus), the etiology of new-onset diabetes in the context of COVID-19 is not known. However, it is likely multifactorial, potentially involving impairments in both glucose disposal and insulin secretion, stress hyperglycemia, the unmasking of pre-existing undiagnosed diabetes, and steroid-induced hyperglycemia [4]. COVID-19 may induce an inflammatory state resembling that which is observed in T2DM which effects may lead to beta cell exhaustion and worsening of diabetes caused by islet hyperstimulation and glucose toxicity as described in Montefusco et al. [11].

A key aspect of SARS-CoV-2 pathogenesis involves its binding to the angiotensin-converting enzyme 2 (ACE2) receptor, which is expressed on the cell membranes of various organs critically involved in metabolic regulation, including the lungs, heart, gastrointestinal tract, liver, kidney, and pancreas [10,11,12,13,14,15]. The virus utilizes its spike protein to engage with the ACE2 receptor, facilitating cellular entry via endocytosis [10,12]. Given the widespread presence of ACE2 receptors in the metabolically active tissues, it is plausible that SARS-CoV2 infection could directly contribute to metabolic disturbances [12,13,14,16,17,18]. This is supported by findings from Washirasaksiri et al., who demonstrated significant long-term metabolic abnormalities, including increased rates of pre-diabetes, diabetes, hyperlipidemia, and indicators of liver inflammation, in previously healthy individuals following even non-severe SARS-CoV-2 infections [12,13,16].

The heightened susceptibility of individuals with pre-existing metabolic conditions such as cardiovascular disease, diabetes, hypertension, obesity, to severe COVID-19 outcomes is well-established. This suggests a potential bidirectional relationship, where metabolic syndrome may significantly influence COVID-19 pathophysiology, and conversely, SARS-CoV-2 infection may precipitate or exacerbate metabolic dysfunction. The shared involvement of the ACE2 receptor in the pathogenesis of hypertension, diabetes and coronavirus infection further supports this hypothesis [12,18,19]. The challenge in confirming this hypothesis is the disentangling of the direct pathophysiological effects from the pandemic itself. The COVID-19 pandemic introduced widespread societal changes and interruptions to normal daily activities, including activities of daily living and diet. Concurrent with the global pandemic, observational studies have reported a concerning rise in the incidence of obesity and type 2 diabetes(T2DM) [13,14,15,16,17,18,19]. Therefore, it has been difficult to distinguish the direct effects of SARS-CoV-2 infection from the metabolic consequences of these population-level lifestyle shifts.

We addressed this by evaluating the risk of new-onset metabolic disorders—including T2DM, hypertension (HTN), hyperlipidemia (HLD), metabolic dysfunction-associated steatotic liver disease (MASLD), and metabolic syndrome (MetS)—among a large, uniquely healthy cohort of U.S. active-duty service members (ADSM). The combination of rigorous physical fitness standards and universal healthcare in this population makes it ideal for a propensity score-matched cohort analysis of post-COVID-19 metabolic disorders.

## 2. Materials and Methods

This investigation employed a propensity score-matched cohort study design, utilizing data from the TRICARE Management Activity’s Military Health System Data Repository (MDR), which tracks all healthcare billing for U.S. uniformed service members and their families. This data includes care received at both civilian and military treatment facilities worldwide, providing access to a population that spans a broad spectrum of socioeconomic backgrounds and geographic locations.

The study cohort composed of ADSM, 17–64 years of age, with data drawn from U.S. military health system healthcare claims from July 2019 to June 2022. Individuals aged 65 and older (the mandatory age of retirement), members of the National Guard and Reserve, and those with no recorded encounters over the previous year were excluded. COVID-19 infections were ascertained from July 2020 to June 2021. A COVID-19 diagnosis was determined using either ICD-10 diagnosis codes (available from April 2020) or positive antigen or polymerase chain reaction (PCR) laboratory test results (with widespread testing available from June 2020). These diagnoses were identified from inpatient and outpatient encounters at civilian or military treatment facilities. The outcomes of interest–T2DM, HTN, HLD, MASLD, and MetS–along with overweight and obesity status, were defined using specific ICD-10 diagnosis codes listed in Table A1. Of note, the definition of MASLD was made in June 2023 replacing the older terms for nonalcoholic fatty liver disease (NASFLD) and nonalcoholic steatohepatitis [20]. For the purposes of this study we used the older ICD-10 codes for NAFLD and NASH as there are no current ICD-10 codes representing MASLD. These ICD codes have previously been substituted for MASLD in other studies [20].

For the generation of the propensity score, clinical data in the year preceding COVID-19 infection was obtained. These data included inpatient and outpatient encounters with ICD-10 diagnosis codes, ICD-10 procedure codes, Healthcare Common Procedure Coding System codes and dispensed outpatient medications. The diagnosis and procedure codes were grouped into Clinical Classifications Software Refined (CCSR) categories using mapping programs from the Agency for Healthcare Research and Quality (AHRQ) Healthcare Cost and Utilization Project: the DXCCSR Mapping Program (vers. 2022.1, AHRQ, 2021), PRCCSR Mapping Program (vers. 2023.1, AHRQ, 2022), and CCS Services Procedures Mapping Program (vers. 2022.1, AHRQ, 2021). Total counts for each category were summed. Pharmacy prescriptions were summed as total days supply of dispensed medication in the previous year and grouped by the American Hospital Formulary Service therapeutic class codes.

Our study matched COVID-19-exposed individuals to unexposed controls in a 1:2 ratio from July 2020 to June 2021. The propensity score model accounted for extensive clinical and pharmacy data from the 12 months before infection, including 1850 CCSR diagnosis, procedural and pharmacy categories to ensure a robust match. Variables with fewer than 100 occurrences, or those present exclusively in either the exposed or control groups, were removed to ensure model convergence. Additional factors such as other health insurance indicators and the percentage of care received in direct care at a military treatment facility were also considered. A logistic regression model was employed to calculate the individual-level probability of COVID-19. Individuals with a record of a COVID-19 infection were matched to non-exposed controls at a 1:2 ratio using greedy matching without replacement. Individuals were matched based on the nearest Euclidean distance of the logit of the propensity score, with calipers set at 0.2 times the standard deviation of the logit of the propensity score, consistent with previously described methods [21]. To ensure robust comparability, individuals within matched groups were required to match exactly on age (in years) and sex.

Once matched, a subject was generally not eligible for propensity score modeling or matching in subsequent months. An exception was made if the individual was initially included as a control but later developed COVID-19; in such cases, these individuals were censored as controls at the time of their infection. To assess the effectiveness of the matching process, the standardized mean difference for each variable included in the model was evaluated both before and after matching. A standardized mean difference below 0.1 was set as the criterion indicating an adequate level of matching to minimize confounding [22,23].

To ensure that only new-onset conditions were captured, we excluded any matched groups where a participant had a prior diagnosis of the outcome of interest. The analysis began 31 days after a COVID-19 diagnosis to avoid bias from increased healthcare use immediately after infection. Individuals were then followed for up to one year to track the development of T2DM, HTN, HLD, MASLD, and MetS. Subjects were censored if they lost healthcare eligibility for two or more consecutive months, or if, in the control group, they received a subsequent COVID-19 diagnosis.

Conditional Cox proportional hazard models were employed to estimate hazard ratios (HRs) for the development of each outcome among the matched groups. Five primary outcomes were evaluated as dependent variables in five separate models. As a sensitivity analysis, we also evaluated a medical condition unlikely to be related to COVID-19, but which is still common within the military population, ingrown toenails (ICD-10 L600). Statistical significance was determined at an overall confidence level of 95% using a Bonferroni correction for multiple comparisons, leading to individual confidence intervals (CI) of (1 − (0.05/5))% = 99% for each of the five outcomes. This adjustment is equivalent to modifying the significance level from alpha = 0.05 to alpha = 0.01 to account for the evaluation of five distinct outcomes. We refer to these adjusted confidence intervals as Bonferroni-corrected 95% confidence intervals. The independent variables considered were COVID-19 exposure and a prior diagnosis of overweight or obesity. The proportional hazards assumption was assessed using Schoenfeld residuals. Where a violation of this assumption was detected, a time-interaction term was incorporated into the Cox regression model to account for time-dependent hazard ratios.

The MatchIt package was used for matching and the smd package for evaluating standardized differences in R [24,25,26]. All other statistical analyses were performed using SAS v9.4 (Cary, NC, USA). The study protocol received review and approval from the Uniformed Services University Institutional Review Board.

## 3. Results

### 3.1. Cohort Matching Process and Overall Demographics

The calculation of the propensity score utilized data from a total of 1,383,169 ADSM. The final propensity score-matched cohort included 103,789 COVID-19 exposed individuals and 207,578 controls, maintaining a precise 1:2 ratio (see Table A2). The median age for both groups was 26 years, with an interquartile range (IQR) of 22 to 33 years, indicating a predominantly young adult population. The cohorts were well-balanced demographically with both groups being predominantly male (81.12% in both cohorts) and composed of similar proportions of different racial/ethnic groups and military ranks. The prevalence of prior overweight or obesity diagnoses was relatively low and nearly identical between groups at 12.97% among exposures and 12.30% among controls, demonstrating the “healthy cohort” nature of this population. After matching, the average propensity scores were nearly identical for both groups (0.006), and visual inspection of the data in Figure A1 confirmed excellent balance across all covariates with no variables in any month having a standardized mean difference greater than or equal to 0.1.

### 3.2. Disease-Specific Cohort Demographics and Incident Rates

Following the exclusion of individuals with a pre-existing diagnosis of the outcome of interest, the sizes of these five cohorts were as follows: T2DM: 102,685 exposures and 205,370 controls; HTN: 93,041 exposures and 186,082 controls; HLD: 95,941 exposures and 191,882 controls; MASLD: 102,632 exposures and 205,264 controls; and MetS: 103,654 exposures and 207,308 controls.

Analysis of the five disease-specific cohorts (see Table A3) revealed that individuals with incident outcomes were consistently older than those without. Males predominated across most conditions, though there were more females who developed metabolic syndrome. Racial disparities were observed, with Black individuals overrepresented in T2DM and HTN and Hispanic individuals in MASLD and MetS. Senior enlisted personnel more commonly developed outcomes. Most notably, prior overweight/obesity was commonly observed among all incident outcomes.

Among those with COVID-19, the incident rates for the respective outcomes were: 211 of 102,685 (0.21%) for T2DM; 1972 of 93,041 (2.12%) for HTN; 1906 of 95,941 (1.99%) for HLD; 475 of 102,632 (0.46%) for MASLD; and 60 of 103,654 (0.06%) for MetS. Cumulative incidence curvesvisually shown in Figure 1.

### 3.3. Survival Analysis Results

The median time in days to a new diagnosis was highly consistent across all outcomes, as shown in Table 1, with ranges of 178–199 days to diagnosis for COVID-19-exposed individuals and 174–197 days to diagnosis for controls.

The analysis revealed that COVID-19 infection was associated with a significantly elevated risk for certain metabolic conditions. The Bonferroni-corrected unadjusted and adjusted hazard ratios, controlling for overweight and obesity status, are presented in Table 2. The proportional hazards assumptions were met for all predictors except for COVID-19 with lipid disorders; consequently, the hazard ratios for this outcome varied with time and are noted accordingly in the table. After adjusting for prior overweight and obesity diagnoses, the following associations remained significant: HTN (aHR 1.09; 99% CI: 1.01, 1.18), HLD (aHR 1.30; 99% CI: 1.10, 1.54; decreased with time) and MASLD (aHR 1.36; 99% CI: 1.15, 1.60). In contrast, there was no statistically significant association between COVID-19 and new diagnoses of T2DM (aHR 0.95; 99% CI: 0.75, 1.21) or MetS (aHR 1.15; 99% CI: 0.70, 1.90). For HLD, at the start of the study (day 0), the hazard ratio for COVID-19 exposures compared to controls is 1.30 (1.10, 1.54) and decreases with time. After 365 days, the hazard ratio becomes 0.92 (0.78, 1.08). For our sensitivity analysis for a condition unlikely to be related to COVID-19, COVID-19 was not associated with ingrown toenails (aHR 1.00; 95% CI: 0.91–1.10).

It is important to note that a prior diagnosis of overweight or obesity was a powerful independent predictor for all outcomes, with adjusted hazard ratios for overweight ranging from 1.29–2.40 and for obesity ranging from 2.15–6.36 depending on the condition (Table 2).

## 4. Discussion

This study provides strong evidence that COVID-19 infection is directly associated with an increased risk of developing HTN, HLD and MASLD in a healthy, young adult population. This association remains significant after adjusting for a prior diagnosis of overweight or obesity and suggests the potential for a direct pathogenic role of SARS-CoV-2. These findings align with other investigations identifying these metabolic conditions as common components of long COVID-19 [12,15,16,27]. Specifically, Akpek et al. and Bieleck et al. demonstrate the increased incidence of HTN following exposure to COVID-19 while noting the potential of several confounders including increasing sedentary lifestyle as a result of the pandemic [28,29]. HLD has also been previously observed to be increased after COVID-19 infection [30,31]. While more severe COVID-19 infection is known to be associated with preexisting MASLD, there are a paucity of studies evaluating the development of MASLD after a COVID-19 infection [32]. One Mendelian randomization (MR) study did not find an association between COVID-19 and non-alcoholic fatty liver disease (the condition now known as MASLD). The discrepancy with our findings is likely related to population variation. MR studies utilize existing genetic databases which are often narrowly geographically and genetically focused whereas our study has a large racially and geographically diverse population [33].

A particularly important finding from this study is the lack of a significant association between COVID-19 infection and new-onset T2DM or MetS. While some previous population studies have linked COVID-19 to higher rates of T2DM, with those with more severe disease having higher risk. Celelja et al. did not find an association with COVID-19 and T2DM using a MR technique [12,34]. These discrepancies likely represent differences in populations and severity of COVID-19. Our findings likely reflect the younger age and generally healthy baseline health of the ADSM cohort. This supports the “healthy cohort hypothesis,” suggesting that COVID-19’s diabetogenic effects may be more pronounced in populations with pre-existing metabolic vulnerabilities, like obesity. The direct connection between COVID-19 and metabolic syndrome has been less studied. However, there are other studies suggesting a connection between COVID-19 and metabolic syndrome through multiple mechanisms, including deregulated immunity, endothelial dysfunction and gut dysbiosis. The lack of association between COVID-19 and MetS in our population may be because of lower numbers or lack of formal diagnosis extended beyond what is seen the medical chart [35].

As previously mentioned, the link likely arises from the virus’s ability to use the ACE2 receptor to trigger inflammation in key metabolic organs, leading to secondary hypertension, dyslipidemia and steatotic liver disease [28,29]. Beyond the connection between COVID-19, the ACE2 receptor and observed metabolic outcomes, there are several other biological mechanisms that may support the plausibility for the observed associations of new-onset HTN, HLD, and MASLD following COVID-19. Generally, lipids have been shown to play an important role in infection clearance and serum low-density lipoprotein (LDL), high-density lipoprotein (HDL) and triglycerides levels all increase in response to acute infection with COVID-19 [36]. The development of dyslipidemia may be linked to the virus’s interaction with lipid rafts—cholesterol-rich domains on the plasma membrane that can facilitate viral entry [37]. The resulting dyslipidemia and accumulation of LDLs can, in turn, induce inflammasome activation and the release of pro-inflammatory cytokines such as IL-1 and IL-6, promoting a systemic inflammatory state that could drive the pathophysiology of both dyslipidemia and steatotic liver disease [38,39]. The significantly elevated hazard of MASLD despite obesity status endorses COVID-19’s role in metabolic liver dysfunction [40]. The comprehensive review by Buchynskyi et al. showcases significant markers of liver damage and gut microbiome disruption following COVID-19 infection leaving patients at significant risk of MASLD [32,41].

A unique finding is that while COVID-19 increased the risk of these metabolic conditions, it did not significantly shorten the median time to diagnosis, which consistently remained around six to seven months for all conditions (Table 1). This suggests that COVID-19 is a risk factor that increases the overall incidence of these diagnoses without altering the underlying latency period. These findings suggest that the observed association was not likely due to testing bias around COVID-19 infection.

Our findings suggest prior COVID-19 infection as an independent risk factor for the development of metabolic disease. The military cohort is generally young, healthy, physically fit and subject to initial rigorous health standards and continued regular fitness and physical exam standards. These findings in this population may shift the paradigm of risk factors for metabolic disease such as age and sedentary lifestyle to be more expansive to include anyone infected with COVID-19. It is notable that the metabolic risk factor could be confounded by pandemic-related lifestyle shifts such as stress levels, healthcare access and new obesity status, but it is difficult to completely disentangle these confounders. Our study did not specifically study the effects of COVID-19 vaccination or long-term effects beyond a year. However, it may be reasonable to consider the possibility that methods to prevent COVID-19 infections may also prevent secondary metabolic dysfunction and cardiovascular disease.

The link between COVID-19 and new-onset HTN, HLD, and MASLD suggests a reemphasis on early identification and treatment. During a post-COVID-19 pandemic period, a renewed commitment to following standard screening guidelines for HTN and HLD is necessary [42,43]. While universal screening for MASLD is not currently standard practice, our findings suggest targeted screening should be considered, using liver enzyme tests in post-COVID-19 patients, especially those who present with other cardiovascular risk factors or have a history of overweight or obesity [44].

The strong association between prior overweight or obesity status along with COVID-19 infection with metabolic disease reinforces that weight management is the cornerstone of metabolic health. Not only does healthy weight lower the baseline risk for severe COVID-19 infections but it also may build resilience against the metabolic dysregulation that can be triggered by viral infections. For ADSM and the general public alike, strategies promoting healthy diet and physical activity are important for long-term disease prevention.

This study has several limitations, including its one-year follow-up period, which may be a short interval to capture all long-term metabolic consequences of COVID-19, such as T2DM and MetS. The study was also unable to fully account for undiagnosed pre-existing conditions or genetic predispositions that could influence metabolic outcomes. A notable limitation is the potential for informative censoring. Individuals in the control group were censored if they received a COVID-19 diagnosis during their follow-up period. Individuals were also censored if they became ineligible for TRICARE for two consecutive months, which could lead to less healthy individuals being censored. Utilizing ICD-10 diagnosis codes is a limitation which could lead to misclassification bias. An example of which is the potential underutilization of syndromic ICD codes such as MetS. The findings may have limited generalizability to the broader U.S. population due to the unique health profile, unique living environments (deployments, barracks), rigorous physical training standards and universal healthcare access of military personnel.

Additionally, we did not adjust for disease severity due to limited measures for assessing this through our study design employing primarily ICD-10 codes and COVID-19 diagnostic laboratory testing alone. However, we attempted to capture some element of the impact of comorbid chronic diseases with the active-duty military cohort.

Despite these limitations, the study benefits from several strengths. It utilizes a large, diverse cohort with universal health care access, required yearly health assessments and evaluation for missed work with illness, ensuring robust data capture. The study also employed rigorous propensity score matching over 12 months, using extensive clinical variables exact matching on age and sex, effectively balancing baseline characteristics, closely emulating randomized trial conditions. By minimizing confounding, the researchers ensured that observed associations were more likely due to COVID-19 exposure rather than pre-existing differences, strengthening the internal validity of the findings.

The study followed subjects for only one-year post-infection. Some metabolic disorders (especially T2DM, MASLD) develop gradually. Future directions of research include extending follow-up to 2–5 years would improve detection of long-term sequelae. We hope to also include data on multiple infections or long COVID diagnoses that could identify cumulative or persistent risk effects.

## 5. Conclusions

The study results demonstrated increased risk post-COVID-19 of developing HTN, HLD and MASLD, independent of diagnosis of overweight or obesity, but did not demonstrate a significant risk of development of T2DM or MetS. These findings, established through rigorous propensity score matching within a uniquely young and healthy military cohort, strengthens the evidence for a direct association between SARS-CoV-2 infection and long-term metabolic consequences. This highlights the need for targeted health monitoring—especially for blood pressure, lipid and liver dysfunction—for those who have previously been infected by the SARS-CoV-2 virus. Additional continued surveillance and targeted research are crucial to fully understand the long-term health consequences of COVID-19 and to inform appropriate preventative and management strategies.

## Figures and Tables

**Figure 1 metabolites-15-00795-f001:**
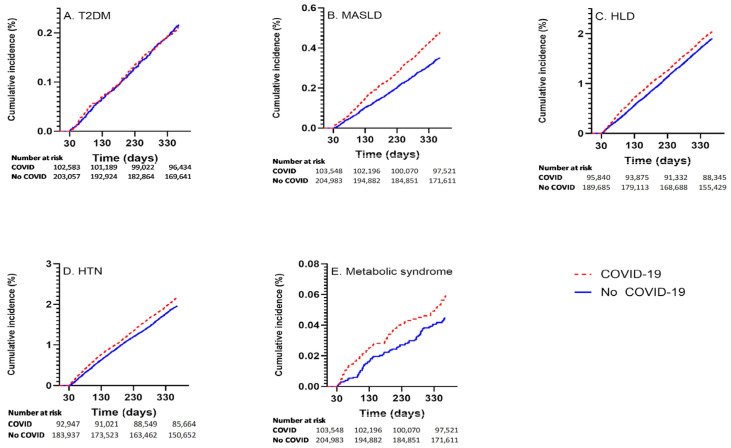
Cumulative incidence curves showing outcome incidence in COVID-19 exposed and unexposed groups for (**A**) T2DM, (**B**) MASLD, (**C**) HLD, (**D**) HTN, (**E**) MetS.

**Table 1 metabolites-15-00795-t001:** Median Days to outcome by COVID-19 exposure.

Outcome	COVID-19 Exposed	COVID-19 Not Exposed
Type 2 diabetes	189 (94, 266)	192 (104, 277)
Hypertension	187 (96, 272)	182 (103, 275)
Hyperlipidemia	180 (98, 273)	191 (110, 276)
MASLD	199 (112, 285)	197 (113, 278)
Metabolic syndrome	178 (77, 269)	174 (106, 279)

**Table 2 metabolites-15-00795-t002:** Adjusted and Unadjusted Hazard Ratio Results for Selected Outcomes: Hazard ratios and Bonferroni-corrected confidence intervals after adjustment for covariates (COVID-19 and prior obesity or overweight diagnosis).

	Type 2 Diabetes		Hypertension		Hyperlipidemia		MASLD		Metabolic Syndrome	
	uHR (CI)	aHR (CI)	uHR (CI)	aHR (CI)	uHR (CI)	aHR (CI)	uHR (CI)	aHR (CI)	uHR (CI)	aHR (CI)
COVID-19	1.00 (0.81, 1.25)	0.95 (0.75, 1.21)	1.11 (1.03, 1.20)	1.09 (1.01, 1.18)	1.32 (1.12, 1.56) ^	1.30 (1.10, 1.54) ^	1.40 (1.19, 1.63)	1.36 (1.15, 1.60)	1.36 (0.87, 2.11)	1.15 (0.70, 1.90)
Prior Overweight Diagnosis	1.44 (0.85, 2.44)	2.40 (1.36, 4.23)	1.13 (0.94, 1.37)	1.29 (1.06, 1.56)	1.35 (1.13, 1.62)	1.54 (1.28, 1.84)	0.97 (0.66, 1.44)	1.37 (0.91, 2.06)	0.88 (0.35, 2.23	2.16 (0.76, 6.19)
Prior Obesity Diagnosis	4.90 (3.42, 7.04)	5.43 (3.74, 7.87)	2.13 (1.86, 2.45)	2.16 (1.89, 2.49)	2.08 (1.82, 2.37)	2.15 (1.88, 2.45)	3.98 (3.03, 5.22)	4.05 (3.07, 5.34)	5.53 (2.79, 10.96)	6.36 (3.02, 13.37)

^ indicates a significant interaction with time that decreases hazard ratio over time, MASLD = Metabolic Dysfunction-Associated Steatotic Liver Disease, uHR = unadjusted hazard ratios, aHR = adjusted hazard ratios, CI = confidence interval.

## Data Availability

All data were accessed from the Military Health System database, which requires a valid data sharing agreement due to the nature of personal health information and cannot be shared without period approval from the U.S. Department of Defense, Defense Health Agency.

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
