# Peer review of "The Metabolic Aftershock: COVID-19 and Metabolic Disease Risk Among U.S. Active-Duty Military Personnel"

_metabolites, 2025, doi:10.3390/metabo15120795_

Round 1

Reviewer 1 Report

Comments and Suggestions for Authors

Dear authors,

The submitted manuscript aims at investigating the association between SARS-CoV-2 infection and the onset of metabolic disorders, namely hypertension (HTN), hyperlipidemia (HLD), metabolic dysfunction-associated steatotic liver disease (MASLD), Type 2 Diabetes Mellitus (T2DM), and Metabolic Syndrome (MetS), among a large cohort of U.S. active-duty service members (ADSM). In this brief report, authors describe an association between increased incidence of HTN, HLD, and MASLD within one year, independent of prior overweight or obesity status, while no significant association was found for T2DM or MetS. These conclusions rely on a solid study design, where a unique and high-quality cohort of individuals has been assessed, with rigorous statistical analysis. Authors conclude that—according to the presented findings—targeted blood pressure, lipid and liver function monitoring is recommended for people who have previously been infected by the SARS-CoV-2 virus. Major limits of the study consist in the limited time of observation and in the lack of an external validation cohort, as detailed in the following list of minor comments:

Minor comments

  1. This study relies on a unique and high-quality cohort of individuals. However, for its intrinsic nature, this also limits the generalizability of findings, calling for the need of an external validation cohort. This should be discussed by authors in the text as appropriate
  2. Beside the “healthy cohort hypothesis”, several additional reasons call for caution when commenting the lack of association observed between SARS-CoV-2 infection and post-COVID-19 T2DM or MetS onset. Among them, (i) the time of follow-up, which may be too short to capture the full spectrum of long-term metabolic consequences, and (ii) the use of ICD-10 diagnosis codes, which may fail to capture MetS or early T2DM if laboratory and physical parameters defining such conditions are not formally coded as the 'syndrome' diagnosis. This should be discussed in the manuscript.
  3. There are tables that are not placed in the appropriate order (e.g. tables 1 and 2), or figures that are not cited at all in the main text (e.g. Figure 1)
  4. Several additional relevant reports may help clarifying the putative post-COVID-19/metabolic disease relationship (e.g. Montefusco L et al, Nat Metab, 2021) and providing underlying mechanistic hints (e.g. Loretelli C et al, JCI Insight, 2021).

Author Response

Dear Reviewer 1,  

We sincerely thank you for the time, expertise and thoughtful feedback provided of our manuscript. We found the ideas and comments helpful in strengthening the rigor and quality of our work. The changes referenced in this document have been incorporated into the manuscript and are expressed in detail below.

Comment 1[This study relies on a unique and high-quality cohort of individuals. However, for its intrinsic nature, this also limits the generalizability of findings, calling for the need of an external validation cohort. This should be discussed by authors in the text as appropriate – covered by new analysis of “pseudo-control”] 

Response 1 [As rightfully stated, this study relies on a unique and high-quality cohort of individuals and its intrinsic nature could potentially limit the generalizability of findings. Given this, we developed a sensitivity analysis by testing an outcome we believed would be both common among the ADSM population and medically unrelated to COVID-19: ingrowing toenail (ICD-10 code L60.0). The hazard ratios were not significant: 1.01 (0.92, 1.11) before adjustment for overweight/obesity and 1.00 (0.91, 1.10) after adjustment.]

Comment 2 [Beside the “healthy cohort hypothesis”, several additional reasons call for caution when commenting the lack of association observed between SARS-CoV-2 infection and post-COVID-19 T2DM or MetS onset. Among them, (i) the time of follow-up, which may be too short to capture the full spectrum of long-term metabolic consequences, and (ii) the use of ICD-10 diagnosis codes, which may fail to capture MetS or early T2DM if laboratory and physical parameters defining such conditions are not formally coded as the 'syndrome' diagnosis. This should be discussed in the manuscript.]

Response 2[We agree with the reviewer’s point. To address this, we have now discussed further limitations to the study including time of follow-up and use of ICD-10 codes.]

Comment 3 [There are tables that are not placed in the appropriate order (e.g. tables 1 and 2), or figures that are not cited at all in the main text (e.g. Figure 1)]

Response 3[Thank you for your comment, we have addressed the order of the tables and the citing of the figure 1.

Comment 4 [Several additional relevant reports may help clarifying the putative post-COVID-19/metabolic disease relationship (e.g. Montefusco L et al, Nat Metab, 2021) and providing underlying mechanistic hints (e.g. Loretelli C et al, JCI Insight, 2021).]

Response 4[The reports as provided have been added and specifically discussed throughout the introduction and discussion.]

Thank you again for your constructive and insightful feedback. We hope the revised version of this manuscript meets your expectations and we look forward to your further review.

Sincerely,

Kyle Sexton MD, MS; on behalf of all authors

Tripler Army Medical Center

Kyle.w.sexton3.mil@health .mil

Reviewer 2 Report

Comments and Suggestions for Authors

Thank you for the opportunity to review this proposed manuscript, which investigates the potential long-term metabolic consequences of SARS-CoV-2 infection, specifically within a unique, large cohort of young and generally healthy U.S. active-duty service members. The findings regarding increased risks for hypertension, hyperlipidemia, and MASLD are noteworthy. However, several major methodological limitations and aspects requiring clarification impact the interpretation and certainty of the conclusions. Here is a list of major and minor areas for improvement that I strongly recommend the authors follow-up if they indeed decide to resubmit the manuscript. 

Major Areas for Improvement

1. The study relies exclusively on ICD-10 codes derived from healthcare claims data to ascertain incident diagnoses of the metabolic outcomes. This approach has significant limitations: while ICD codes are useful for tracking diagnoses, their reliability for identifying new-onset (incident) conditions can be variable. Conditions like MASLD and MetS are often underdiagnosed, inconsistently coded, or only identified during evaluations for other reasons, potentially leading to misclassification bias regarding the timing and true incidence of the disease. The manuscript notes the use of older NAFLD/NASH codes for MASLD due to terminology lag, which adds another layer of complexity. I think that the authors need to address this significant issue. If that is not possible, this is a significant limitation to the study.    2. The study acknowledges the possibility of testing bias around the time of COVID-19 diagnosis but follows participants starting 31 days post-infection. However, individuals recovering from COVID-19 might have increased healthcare interactions during the subsequent year compared to controls, potentially leading to a higher likelihood of detecting pre-existing but undiagnosed metabolic conditions, or diagnosing new conditions earlier than they might otherwise have been found.   3. While the propensity score matching is extensive and appears to achieve good balance on a vast number of measured covariates, the possibility of unmeasured confounding remains a significant limitation inherent to observational studies. Let me be more precise. The ADSM population experiences unique exposures and stressors not typically found in civilian populations (for example, deployment history, specific military occupational specialties involving high physical or psychological stress, unique living environments like barracks, rigorous but variable physical training regimens). These factors could independently influence both the risk of COVID-19 infection/severity and the development of metabolic disorders, and may not be adequately captured by standard healthcare claims data used for matching. The authors need to look into this specific issues to the studied population.    4. Although the study aims to isolate the effect of infection from general pandemic-related lifestyle shifts, it's difficult to completely disentangle these. Unmeasured differences in diet, physical activity changes, stress levels, or healthcare-seeking behavior between infected and non-infected individuals during the follow-up year could still confound the results, even if baseline characteristics were well-matched.   5. The rationale for and potential implications of adjusting for overweight and obesity status in the Cox proportional hazards models after propensity score matching needs further clarification. PSM aims to balance baseline covariates between exposed and control groups. If the matching successfully balanced baseline health status, including factors highly correlated with overweight/obesity, further adjusting for these specific diagnoses in the outcome model might be redundant or could potentially introduce bias, especially if weight status is considered part of the causal pathway or a mediator rather than purely a confounder. The strong independent association found for overweight/obesity highlights its importance, but its inclusion post-matching requires careful justification.   6. I think that the manuscript would benefit from revisiting the references included in the introduction and discussion. Please cite  PMID 34173917.     Minor Areas for Improvement   1. The use of Bonferroni correction is a valid but conservative approach to adjust for multiple comparisons. This stringency might contribute to the non-significant findings for Type 2 Diabetes Mellitus (T2DM) and MetS, especially if the true effect sizes are modest. A brief acknowledgment of the potential impact of this conservative correction could be useful.   2. The results note that the proportional hazards assumption was violated for the association between COVID-19 and HLD, necessitating a time-interaction term. However, the manuscript only reports a single adjusted Hazard Ratio (aHR) of 1.30. It should clarify how this HR should be interpreted in the context of time-dependency (e.g., is it an average HR, or the HR at a specific time point?) and ideally provide more information on how the hazard changes over the one-year follow-up period.   3. The study treats COVID-19 exposure as a binary variable. The severity of the initial infection (e.g., asymptomatic, mild, moderate, severe/hospitalized) is a likely significant determinant of post-acute sequelae, including metabolic dysfunction. The lack of stratification or adjustment based on severity limits the granularity of the findings.   4. As acknowledged by the authors, the findings from this unique ADSM cohort (young, predominantly male, generally healthy, with universal healthcare access) may not be directly generalizable to older, more diverse civilian populations with different baseline health profiles and healthcare access patterns.

Author Response

Dear Reviewer 2,  

We sincerely thank you for the time, expertise and thoughtful feedback provided of our manuscript. We found the ideas and detailed comments helpful in strengthening the rigor and quality of our work. The changes referenced in this document have been incorporated into the manuscript and are expressed in detail below.

Major areas of improvement:

  1. We agree with the reviewer’s point. To address this, we have now discussed further limitations to the study including time of follow-up and use of ICD-10 codes. We also addressed the use of older NAFLD/NASH codes for MASLD due to terminology lag, with a new sited reference within the methods which demonstrates it use and significance.    
  2. While we do acknowledge that this has the potential for bias, in our study when looking at days 31-365 post follow-up, those with COVID-19 had a median of 11 visits/year (IQR 6-22) while those without COVID-19 had a median of 10 visits/per (IQR 5-19), demonstrating similarities between these two groups, but it is worth noting that it marginally higher in those with COVID-19. Additional visits may be a result of increased healthcare seeking behavior, but could also be related to the repeated care required for diagnosis and subsequent follow-up care needed for treatment of these newly diagnosed conditions. 
  3. We agree that the ADSM population experiences unique exposures and stressors not typically found in civilian populations and have now discussed in limitations to the study.
  4. We agree that it is difficult to isolate the effect of infection from general pandemic-related lifestyle shifts. To address this, we have now discussed further limitations to the study.
  5. While the inclusion might seem conservative, it had a minor impact on the hazard ratios, and didn't change the statistical significance of any of our models or the conclusions reached. Leaving out such an important predictor of these outcomes would have also garnered concern and would have been viewed as problematic as weight status is highly correlated with the conditions we are studying. Overweight and obese past year history was only one of 1800 variables that contributed to the propensity score, so likely only partially contributed to the propensity score generation. In this case we are conservative in our methods. .
  6. The reports as provided have been added and references have been discussed throughout the introduction and discussion.

Minor areas of improvement:

  1. Bonferroni correction did not change the significance of any of our COVID-19 relationships with the outcomes. While we didn't include the p-values in the manuscript, the p-value for T1DM was 0.61 in the adjusted model, well above both the standard alpha of 0.05 and the Bonferroni corrected alpha of 0.01. The Bonferroni corrections only applied to the confidence intervals and had no impact on the effect size hazard ratios themselves. 
  2. We thank you for this feedback. We added language to the manuscript (located on page 5) to describe how to interpret the hazard ratio for HLD and how it changes over time.
  3. The lack of stratification or adjustment based on severity has been added to our limitations of our study. 
  4. We added language to the limitations section (page 8) to clarify that the generalizability of our findings may be limited to the population we studied (and why) and that pandemic-related lifestyle changes could also have an effect (page 7).

Thank you again for your constructive and insightful feedback. We hope the revised version of this manuscript meets your expectations and we look forward to your further review.

Sincerely,

Kyle Sexton MD, MS; on behalf of all authors

Tripler Army Medical Center

Kyle.w.sexton3.mil@health .mil

Reviewer 3 Report

Comments and Suggestions for Authors

Based on the uploaded article here’s an analysis of the original and relevant contributions and the specific gap the study addresses. Older or comorbid populations (e.g., individuals with obesity, diabetes, or cardiovascular disease). Civilian datasets affected by confounding lifestyle changes during the pandemic (diet, activity level, stress). Based on a full review of the article here are targeted recommendations regarding methodological improvements and further controls the authors should consider:

  1. The study followed subjects for only one year post-infection. Some metabolic disorders (especially T2DM, MASLD) develop gradually; extending follow-up to 2–5 years would improve detection of long-term sequelae.
  2. The authors acknowledge pandemic-related lifestyle changes. Including BMI change, physical fitness scores, or activity/dietary data from military fitness assessments could help disentangle viral from behavioral effects.
  3. Including unrelated outcomes (fracture, skin infection) could help verify that associations are not driven by differential healthcare use.
  4. Including data on multiple infections or long COVID diagnoses could identify cumulative or persistent risk effects.

In summary, while the study is methodologically solid with its prospective, controlled, and randomized design, integrating these improvements would significantly enhance.

Author Response

Dear Reviewer 3,  

We sincerely thank you for the time, expertise and thoughtful feedback provided of our manuscript. We found the ideas and comments helpful in strengthening the rigor and quality of our work. The changes referenced in this document have been incorporated into the manuscript and are expressed in detail below.

  1. We agree with the reviewer’s point. To address this, we have now discussed further limitations to the study including time of follow-up and discussed further directions of future research.
  2. BMI measurements could be used however we were concerned about the accuracy of BMI as a anthropometric indicator of overweight status among individuals whose occupation may lead to higher lean body mass. We felt a diagnosis would be a more specific indicator of a medical diagnosed weight status. Military fitness testing scores were not available in our data source.
  3. As rightfully stated, this study relies on a unique and high-quality cohort of individuals and its intrinsic nature could potentially limit the generalizability of findings. Given this, we developed completed a sensitivity analysis by testing an outcome we believed would be both common among the ADSM population and medically unrelated to COVID-19: ingrowing toenail (ICD-10 code L60.0). The hazard ratios were not significant: 1.01 (0.92, 1.11) before adjustment for overweight/obesity and 1.00 (0.91, 1.10) after adjustment. We have added this to the manuscript.
  4. We agree with the reviewer’s point and discussed further directions of future research.

Thank you again for your constructive and insightful feedback. We hope the revised version of this manuscript meets your expectations and we look forward to your further review.

Sincerely,

Kyle Sexton MD, MS; on behalf of all authors

Tripler Army Medical Center

Kyle.w.sexton3.mil@health .mil